# How Well Are Pharmacists Represented in National Institutes of Health R01 Funding to United States Schools of Pharmacy?

**DOI:** 10.3390/pharmacy10060165

**Published:** 2022-11-30

**Authors:** Duong Nguyen, Ashley R. Selby, Ronald G. Hall

**Affiliations:** Department of Pharmacy Practice, Jerry H. Hodge School of Pharmacy, Texas Tech University Health Sciences Center, Dallas, TX 75235, USA

**Keywords:** pharmacist, grants, National Institutes of Health, representativeness

## Abstract

Pharmacists are essential healthcare providers but historically are not well represented as principal investigators (PIs) of R01 grants by the United States (US) National Institutes of Health (NIH). Pharmacy organizations have taken steps to provide pharmacists with research training to improve their chances of achieving PI status. We conducted a retrospective cohort study using data from the NIH RePORTER website about R01 grants awarded to PIs affiliated with US Schools of Pharmacy (SOPs) for the fiscal years 2005–2019. Information regarding professional degrees was supplemented using data from the PIs’ institutional website profiles and other internet-based sources. Only doctorate degrees obtained within the US were included for clinically related degrees. Data regarding more than one year of funding for the same project, equipment supplements, and diversity supplements were excluded to focus on unique projects in year one of funding. PhDs were the primary unique PIs of R01 grants at US SOPs (>90%). Pharmacist representation as unique PIs increased over the 15 years but was still only 10.1% for the years 2015–2019. There was a higher percentage of female pharmacists as unique PIs than female non-pharmacists. Pharmacists are currently underrepresented as unique PIs for NIH R01 grants. This conclusion is limited by not knowing how many pharmacist R01 applications were submitted.

## 1. Introduction

Pharmacists are viewed as the most accessible healthcare professionals but have historically been limited in conducting National Institutes of Health-funded research as principal investigators (PIs) [1]. The American Association of Colleges of Pharmacy and the American College of Clinical Pharmacy have implemented research development tools to help increase the number of pharmacist PIs for National Institutes of Health-funded research [2,3]. In addition, the National Institutes of Health has funded KL2 programs aiming to develop Clinical Research Scholars as the next generation of clinical and translational research leaders. This multidisciplinary program includes pharmacists as one of the disciplines trained by this funding mechanism.

The percentage of new National Institutes of Health R01 grant awards to pharmacists as PIs within United States Schools of Pharmacy over time has not been evaluated, to our knowledge. Therefore, it is essential to know whether the initiatives to develop pharmacists into leading clinical researchers are resulting in an increased percentage of R01 awards from the National Institutes of Health to pharmacists within Schools of Pharmacy.

Therefore, we collected data from National Institutes of Health Research Portfolio Online Reporting Tools Expenditures and Results (RePORTER) for all R01 grant awards to United States Schools of Pharmacy from the fiscal years (FYs) 2005 to 2019. We sought to determine the percentage of all National Institutes of Health R01 grants awarded to United States Schools of Pharmacy where a pharmacist was a PI. In addition, we aimed to evaluate the trend of the percentage of new National Institutes of Health R01 grant awards to United States Schools of Pharmacy with pharmacists as PIs over time to determine if pharmacists are becoming increasingly competitive for these awards.

## 2. Materials and Methods

This retrospective cohort was developed using the National Institutes of Health RePORTER website. We queried data regarding PIs affiliated with United States Schools of Pharmacy who were awarded R01 grants from the fiscal years 2005 to 2019 from any National Institutes of Health institute or center [4]. Data elements that were exported into a Microsoft Excel file from these queries included the submission title, lead and other PIs’ name(s), which United States School of Pharmacy the lead PI was associated with, the location of the United States School of Pharmacy, years of funding and the amount per year, and the awarding institute or center. Institutional Review Board approval was not required since these data are publicly available.

The primary source of information regarding each United States School of Pharmacy PI’s doctoral and/or clinical degrees was the institutional faculty website profile for each PI. When necessary, further evaluation of other sources was conducted, including the PI’s LinkedIn profiles, lab website(s) for the PI, as well as any news or institutional articles regarding the PI. We included several degrees in the description of the PIs. Data were specifically collected for research doctoral degrees, Doctor of Philosophy (PhD), as well as a number of clinical degrees including for doctors (MD), pharmacists (Bachelor of Science in Pharmacy or PharmD), and public health professionals (MPH). Only clinical degrees (e.g., MD and pharmacy degrees) obtained from United States institutions were included.

Data for the first funding year of each project were included. We excluded equipment and diversity supplements to identify the total number of unique projects in their first year of funding. Descriptive data were reported as counts, percentages, or means. Nominal data were analyzed using a chi-square test. Continuous data were analyzed using a t-test. An alpha of 0.05 was used to evaluate statistical significance. All statistical analyses were conducted with Stata version 15.1 (StataCorp LLC, College Station, TX, USA).

## 3. Results

Nine hundred seventy-eight new National Institutes of Health R01 grants were awarded to 644 lead PIs associated with United States Schools of Pharmacy for the FYs 2005–2019. The breakdown of the degree types for the unique lead PIs is shown in Table 1.

Nine out of ten (*n* = 581, 90%) lead PIs had a PhD as their only doctoral or clinical degree. Twenty-six PIs had a clinical doctorate only. Twenty-five pharmacists also had other degrees, as did seven physicians. The percentage of pharmacists as unique investigators with a new award increased each five-year period, but this increase did not reach statistical significance (*p* = 0.09) (Figure 1).

Those with PhDs, whether as a sole degree or in combination with another degree, were awarded National Institutes of Health R01 grants for 935 projects from 2005 to 2019 as lead PIs (96%). The mean year one total costs were higher for pharmacists compared to other unique PIs within United States Schools of Pharmacy from 2005 to 2019 (USD 490,077 vs. USD 367,035, *p* < 0.001). Unique pharmacist PIs were most commonly funded by the National Institute of Allergy and Infectious Diseases (NIAID) (29%), followed by the National Institute of General Medical Sciences (NIGMS) (19%), and the National Institute of Child Health and Human Development (10%). Unique non-pharmacist PIs were most commonly funded by the National Cancer Institute (NCI) (22%), NIGMS (20%), and NIAID (9%).

Twenty-seven percent of unique PIs were female. The percentage remained between 25 and 30% regardless of the time period (*p* = 0.81). There was a significantly higher percentage of female pharmacists as unique PIs than non-pharmacists (52 vs. 25%, *p* < 0.001). Females were most likely to be funded by the NIGMS (20%), NCI (14%), and NIAID (11%).

## 4. Discussion

Pharmacists are not equitably represented within United States Schools of Pharmacy as lead PIs for National Institutes of Health-funded R01 grants as they compose approximately half of their faculty members [5]. From 2005 to 2019, pharmacists represented 8% of unique PIs for National Institutes of Health-funded R01 grants within US Schools of Pharmacy. While the percentage improvement over each five-year period is encouraging, the raw number of unique PIs still only being 21 for the years 2015–2019 is a sobering reminder of how much work is yet to be done to have research issues that impact contemporary pharmacy practice adequately represented in funding from the National Institutes of Health. The vast majority of unique PIs who were the recipients of National Institutes of Health R01 grants had a PhD, including approximately half of the funded pharmacists. Some would use this statistic as an argument to require PhD training of clinical scientists, but we do not agree with this perspective as the sole solution to increasing the number of pharmacists as lead PIs on NIH R01 grants. A one-size-fits-all PhD requirement of pharmacist scientists ignores the diversity of training programs available to these individuals, including research fellowships and Master’s programs that are focused on clinical and translational research. In addition, there are fewer graduate programs within US Schools of Pharmacy to help train pharmacist scientists in a formal PhD program. Admittedly, not all pharmacist faculty members have sufficient research training to successfully compete for NIH R01 grants as a PI. On the other hand, there are PhD faculty members who are solely on a teaching track and not responsible for pursuing research funding as part of their duties. A positive note is that females represent about half of the pharmacists who were unique PIs.

In 2010, the American College of Clinical Pharmacy provided an update about clinical pharmacists as PIs [1]. They found that five pharmacists received National Institutes of Health funding in 1998, and twenty-four had in 2007. They also noted that only 1.15% of practice faculty members have a National Institutes of Health grant. Their search of clinicaltrials.gov on 6 April 2009, found 43 pharmacists in the role of PI (*n* = 36) or joint PI (*n* = 7). These numbers are higher than what we reported for several potential reasons. We only included R01 funding and the first R01 grant during 2005–2019 when describing the number of unique PIs. A 2013 commentary also noted that some areas of medicine have only a few prominent clinical pharmacy scientist researchers [6]. They suggested this issue is likely multi-factorial, including the lack of a critical mass of mentors and the insufficient number of research training programs (fellowships or graduate programs) as prominent factors. These statements need to be challenged in today’s interdisciplinary environment. The responsibility and opportunity of training pharmacists as clinical researchers is no longer solely the responsibility of existing pharmacist researchers. The KL2 programs funded by the National Institutes of Health are an excellent example of this fact. Furthermore, one could argue that when more disciplines are involved in the training of a future researcher, the perspective of the trainee is enlarged and he/she will have a greater appreciation for working with other disciplines in the future. That being said, some United States Schools of Pharmacy are not part of a Health Science Center, which makes having an onsite multidisciplinary team more difficult. Another option, which was highlighted by the COVID-19 pandemic, is the feasibility of virtual mentoring using videoconferencing for meetings and presentations between the trainee and the mentorship team. Therefore, it is possible for some of the expertise needed to support trainees to be available at remote locations.

Receiving a National Institutes of Health grant as a PI is a significant step towards independence as a researcher. However, the infrequent awarding of these grants to pharmacists as PIs limits their ability to be recognized as clinical scientists. Academic institutions often place National Institutes of Health funding at a premium due to higher indirect cost rates. The perceived inability of pharmacists to compete for National Institutes of Health R01 funding has several downstream implications, including pharmacists’ ability to compete for larger startup packages from Schools of Pharmacy as well as the distribution of effort allowed for research activities. Furthermore, basic science researchers refer to the importance of core facilities as part of an institution’s research infrastructure. There is rarely consideration of the core facilities needed to conduct robust clinical research, further burdening young clinical investigators. The lack of National Institutes of Health funding for pharmacists as clinical/translational scientists may also limit upward mobility as a researcher and/or administrator, as these can require a National Institutes of Health track record. In addition, pharmacists have a unique perspective on aspects of medicine that may be understudied with the current funding strategies.

The relative lack of pharmacist perspectives within National Institutes of Health R01-funded research has many potential impacts on society as well. From a cost perspective, there were USD 348.4 billion spent in the United States on prescription drugs in 2020 [7]. Research focused on penicillin allergy delabeling as an example of how pharmacists and other clinical scientists have determined that the presence of a penicillin allergy in a patient’s medical chart is linked to suboptimal treatment choices that lead to increased costs and worse outcomes [8,9,10]. Twenty years ago, the Institute of Medicine Report entitled “To Err is Human” stated that an estimated 98,000 people die per year from medication errors in a hospital setting [11]. The Agency for Healthcare Research and Quality notes that one of the central responsibilities of pharmacists is to ensure safe medication use [12]. Part of optimal medication use includes the preferential provision of guideline-recommended therapies when appropriate. The addition of a pharmacist to primary care teams improved the use of guideline-recommended antiplatelet agents [13]. In summary, an increased focus on areas of medicine that are areas of expertise for pharmacists can lead to increased prescribing of guideline-recommended therapies as well as increased medication adherence, safety, and effectiveness, while decreasing the cost of care.

This commentary is not an attempt to devalue the valuable work of our basic science colleagues at United States Schools of Pharmacy. It also has not intended to state that most pharmacist faculty members have adequate basic sciences training to compete for NIH R01 grants within these specialty areas. That being said, “NIH’s mission is to seek fundamental knowledge about the nature and behavior of living systems and the application of that knowledge to enhance health, lengthen life, and reduce illness and disability” [14]. We feel that the NIH is not paying adequate attention to pharmacists’ role in the application portion of this mission. No one understands medication use in a clinical setting better than a pharmacist. Pharmacists who have additional training in areas such as epidemiology, economics, and implementation science should be key partners in helping to move the nation’s health forward. Instead of considering pharmacists’ strengths as apples to basic scientists’ oranges, the NIH needs all of us to create a beautiful fruit basket of evidence for society to enjoy.

This study is a retrospective analysis of a publicly available database of successfully funded National Institutes of Health R01 grants awarded to United States Schools of Pharmacy. Therefore, we do not know how many unfunded applications were submitted. We also do not know how many pharmacists were federally funded by non-R01 mechanisms or who received R01s outside of Schools of Pharmacy. Data regarding degrees and sex were not provided as part of the National Institutes of Health RePORTER database, so our web searches for this information could have possibly led to incomplete or inaccurate information. Our definition of a unique PI may have misclassified individuals who had received a previous R01 before 2005. Our decision to not include clinical degrees outside of the United States created a bias that underrepresented pharmacists in the count of unique PIs. However, we believe this is a more practical approach to evaluating individuals who can incorporate contemporary United States pharmacy practice into their research. Additionally, we did not have access to the distribution of effort for the unique PIs included in this analysis. Along these lines, there is a great degree of variation in the commitment to research between United States Schools of Pharmacy and that which is able to be seen in the amount of research funding received by each institution. Furthermore, since we only collected data from the NIH RePORTER database on United States Schools of Pharmacy, we do not have any context of how these findings compare to other disciplines at our healthcare schools (i.e., Schools of Medicine) in the United States.

## 5. Conclusions

Pharmacists are currently underrepresented as unique PIs for National Institutes of Health R01 grants. This conclusion is limited by not including data outside Schools of Pharmacy and not knowing how many pharmacist R01 applications were submitted. Since National Institutes of Health R01 awards underrepresent pharmacists, further study of factors associated with the success of pharmacists as PIs on R01s is needed in collaboration with the National Institutes of Health. More data about the total number of pharmacist R01 applications and factors associated with pharmacists submitting a National Institutes of Health proposal are also required.

## Figures and Tables

**Figure 1 pharmacy-10-00165-f001:**
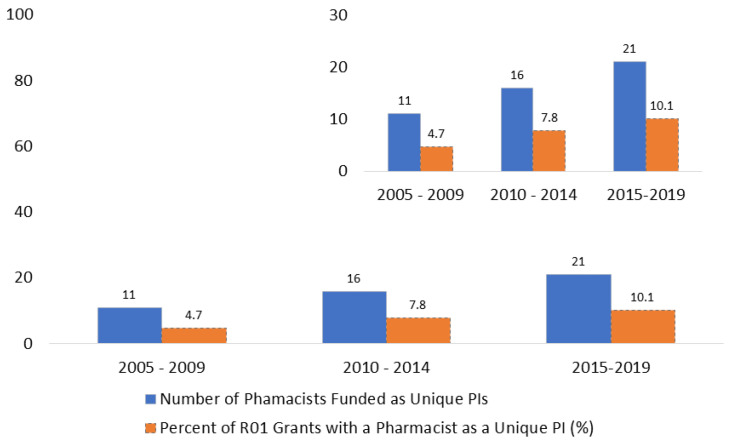
Trends over time in Pharmacists as Unique Principal Investigators (PIs) for R01 grants at United States Schools of Pharmacy (inset graph also presented).

**Table 1 pharmacy-10-00165-t001:** Degrees of Unique Principal Investigators Awarded a R01 at United States Schools of Pharmacy.

Degree(s)	Number	Percent (%)
All	644	100
PhD only	581	90.2
Pharmacist only	23	3.6
Pharmacist/PhD	20	3.1
Pharmacist/MPH	2	0.3
Pharmacist/MPH/PhD	3	0.5
MD only	3	0.5
MD/PhD	6	0.9
MD/MPH/PhD	1	0.2
MPH/PhD	1	0.3
MPH only	2	0.3
Clinical doctorate outside the United States	2	0.3

## Data Availability

Data available in a publicly accessible repository that does not issue DOIs. Publicly available datasets were analyzed in this study. These data can be found here: https://reporter.nih.gov/ (accessed on 7 June 2022).

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
