# Peer review of "How Well Are Pharmacists Represented in National Institutes of Health R01 Funding to United States Schools of Pharmacy?"

_pharmacy, 2022, doi:10.3390/pharmacy10060165_

Round 1

Reviewer 1 Report

I am a non US Pharmacist and read your paper with great interest. Your findings and conclusion would be in line with my thinking and understanding and it is good to see an evaluation to describe these thoughts. However, (perhaps as not a US pharmacist) it wasn't totally clear what was being evaluated against what. I therefore believe the introduction needs to be a little clearer about the relevance of what you are looking at and why you have chosen to look at it the way you have.

The intro keeps talking about the % of pharmacists, but it is not totally clear what the denominator is and the relevance. Assuming it is vs other health care professionals then is there a caveat of what % of all HCPs are pharmacists. 

Does the NIofH only do grants for clinical studies and thus what does the pharmacy picture of workforce look like: clinical pharmacists vs more technical / scientific pharmacists. Would community pharmacy generally go for these sorts of grants.

I fully believe we need more pharmacists involved in research and leading research, but does that mean they need to be the PI?

The key question to me is whether they are not applying or not getting these grants and I think this needs a bit more discussion. Whilst we might agree that more pharmacists should be leading research is that not happening because we are not training them or because as a profession most actually don't want to?

I enjoyed reading this, but believe it can be strengthened by making it clearer with the issues more defined.

Reviewer 2 Report

Thank you for the opportunity to review your paper. Generally, this is a well written paper and i have only a few comments: 

1. under the results, can you clarify the demographic information provided? Line 79 indicates there were 9 of 10 lead PI with PhDs, in additiona to 23 with only a pharmacy degree. The next sentence in line 81 however provides additional information that seems contradictory. I will suggest you rephrase the wording of this section to provide further clarity for the reader. 

2. Please correct typos on the horizontal axis legend in Figure 1. 

Best wishes, 

Reviewer 3 Report

Thank you for the opportunity to read and review this candidate for publication. The authors made a significant effort in this work. Overall the data seem to be reported accurately. The primary take-away from this reviewer is that pharmacist PI is a rare breed and that most of the research conducted in Schools of Pharmacy is the work of the PhD faculty.  This finding seems consistent with this reviewer's experience. The authors also provide examples when pharmacist participation on a research team is of great value (Ref 9-13). That said, good reasons for participation of a discipline is not sufficient for stating that they should be PIs.

 However, there are organizational reasons for this finding that the raw reporting of the data overlooks. As a reviewer, I was hopeful that the manuscript would consider these constraints and would provide some guidance in correcting the perceived "mis-allocation" of pharmacists as PIs. Traditionally Doctor of Pharmacy faculty are actively engaged in clinical instruction (directed at the comment that more than half of SOP faculty are pharmacists). Therefore, between clinical and teaching commitments there is very little time for the effort to be PIs on R01 research grants.  The highest number of pharmacists awarded R01 grants were pharmacist/PhDs. The authors' commented that "Some would use this statistic as an argument to require PhD training of clinical scientists, but we do not agree with this perspective". As a commentary, it would have been enlightening why the authors disagreed with this alternative strategy. However, their comment ended on a simple "we disagree" without a reason for their disagreement when others have earlier discussed this topic (Ref 6) with their rationale and suggestions. This reviewer's primary concern is along this line, as well.  The authors accurately point out the rarity of a pharmacist PI and bemoan this fact. However, it seems to miss those potential legitimate reasons. Just saying that there is this "fact" without participating in the correction just seems a little hollow. 

Round 2

Reviewer 3 Report

Lines 87-93: The numbers of persons with the degree does not need to be within the parentheses because it is in the table.

Figure 1 seems to have the same bar charts twice. If they should be there, then the difference between the two needs to be clarified for the reader.

Lines 125 - 132 only partially cures this Reviewer's primary objection to acceptance of Draft 1. It may have been this Reviewer's shortcoming in stating the comment, but the authors seemed to miss the point in their response.  The authors stated "we disagree with this perspective".  Okay, as the authors they have that right. However, WHY do they disagree and how does the data they presented support their point and not the point(s) of their counterparts? This is the point of a commentary.

In the response to Reviewer #1, the authors state "we are unclear of the reviewer's intent". I would like to suggest an interpretation. Many of the PhD only grant awards are likely for pharmaceutical sciences on topics of pharmacology, medicinal chemistry, etc and not clinical research. PharmDs are likely not well trained in the basic sciences and are less likely to be qualified for R01 grants in those disciplines.  So, are the author's comparing apples to apples or apples to oranges (e.g., pharmaceutical sciences to clinical research)?  If so, is the interpretation of the author(s) the same or correct?
